# BifA Triggers Phosphorylation of Ezrin to Benefit *Streptococcus equi* subsp. *zooepidemicus* Survival from Neutrophils Killing

**DOI:** 10.3390/biomedicines10050932

**Published:** 2022-04-19

**Authors:** Fei Pan, Jie Peng, Dandan Yu, Lianyue Li, Hongjie Fan, Zhe Ma

**Affiliations:** 1MOE Joint International Research Laboratory of Animal Health and Food Safety, College of Veterinary Medicine, Nanjing Agricultural University, Nanjing 210095, China; panfei@stu.njau.edu.cn (F.P.); pengj@gsau.edu.cn (J.P.); veterinary2022@163.com (D.Y.); li_lianyue@outlook.com (L.L.); fhj@njau.edu.cn (H.F.); 2Ministry of Agriculture Key Laboratory of Animal Bacteriology, Nanjing 210095, China; 3College of Veterinary Medicine, Gansu Agricultural University, Lanzhou 730070, China; 4Jiangsu Co-Innovation Center for Prevention and Control of Important Animal Infectious Diseases and Zoonoses, Yangzhou 225009, China

**Keywords:** *Streptococcus equi* subsp. *zooepidemicus*, *bifA*, neutrophils, ezrin

## Abstract

*Streptococcus equi* subsp. *zooepidemicus* (SEZ) ATCC35246 can invade the brain and cause severe neutrophils infiltration in brain tissue. This microorganism can survive and reproduce to an extremely high CFU burden (10^8^–10^9^/organ) under stressful neutrophils infiltration circumstances. The aim of this research is to explore the mechanism of the SEZ hypervirulent strain with its specific *bifA* gene which avoids being eliminated by neutrophils in the brain. We isolated the primary mouse neutrophils to treat SEZ WT and *bifA* gene defective (ΔBif) strains. The ΔBif strain had a weakened function of defending against neutrophils killing in vitro. The interaction between BifA and ezrin proteins in neutrophils were identified by co-IP and immunoblot. In neutrophils, the BifA interacts with ezrin and triggers the phosphorylation of ezrin at its Thr567 site in a PKC-dependent manner, then the excessive elevation of phosphorylated-ezrin recruits Dbl and activates Rac1. Since the Rac1 is closely relevant to several critical cellular functions, its abnormal activation will lead to neutrophils dysfunction and benefit to SEZ survival from neutrophils killing. Our findings reveal a novel consequence of BifA and ERM family protein (for ezrin, radixin, moesin) interaction, which happens between BifA and ezrin in neutrophils and contributes to SEZ survival in the brain. BifA should be considered as a potential target for drug development to prevent SEZ infection.

## 1. Introduction

*Streptococcus equi* subsp. *zooepidemicus* (SEZ) is a Lancefield Group C opportunistic pathogen. The hypervirulent SEZ strains are the major pathogens that cause swine bacterial meningitis and acute death, such as ATCC35246, OH-71905, and TN-74097. The SEZ ATCC35246 has caused a severe outbreak in the Chinese pig feeding industry in the 1970s, and OH-71905 and TN-74097 were reported as pathogens of high mortality in the North American pig industry in 2019 [1,2]. The genomes of these hypervirulent strains from pigs are highly homologous (>99.7%), and all encode a specific Fic (filamentation induced by cyclic AMP) domain protein which is often absent in other isolates from humans or other animals [2,3,4,5]. The consensus nine amino acids core (HxFx(D/E)(A/G)N(K/G)R) is the characteristic of Fic domain. Proteins with this domain are usually present in pathogens as virulent factor and can manipulate host signaling pathways via covalent modification of target proteins [6]. In our previous research, we proved that the SEZ Fic domain-containing protein can be released to the environment and reach as high as ~18 μg/mL in culture supernatants; however, the secretion route is still unknown. This protein was named as BifA due to its brain invasion function of disrupting the blood-brain barrier integrity and activating ERM family proteins moesin in brain microvascular endothelial cells [7]. ERMs expressed in a developmental and tissue-specific manner; the divergent functions of ERMs are tailored to specific cell types, including lymphocyte and leukocyte [8,9,10]. Due to the wide spread of ERMs in different cell types and their vital role in cellular function, these proteins could be ideal targets for pathogens to attack in their infectious strategy [11,12].

As the most abundant leukocytes in the circulation, neutrophils compose the first line of defense in the innate immune system via the capture and elimination of invaded pathogens. Neutrophils infiltration happens during acute inflammatory response against bacterial infection [13]. Although most bacteria cannot survive from neutrophils killing, SEZ ATCC35246 has a unique capacity to circumvent the destruction of neutrophils to a certain extent [14]. The mechanism of this hypervirulent strain to avoid being eliminated by neutrophils in the brain is still unexplored.

Ezrin links the cortical cytoskeleton to the plasma membrane and plays a role in limiting rapid cell shape change in neutrophils [15]. The phagocytosis and motility of neutrophils correlate to ezrin phosphorylation [16]. Though the ERMs share the striking amino acid identity with two conserved domains (FERM and C-ERMAD domains), functional and structural diversity remains. For example, ezrin and moesin have distinguished sensitivity to calpain, and ezrin has its specified phosphorylated Tyr residues while moesin has phosphorylated Thr residues only [17,18]. The interaction between BifA and moesin has been identified in endothelium cells, raising a question about the possibility of BifA function at ezrin phosphorylation regulation in other cell types. Phosphorylated ezrin (p-ezrin) associates with Dbl through its NH2-terminal domain and causes Rho activation [19,20]. The Rho-ROCK signal system involves neutrophils migration, the consequences of Rho-ROCK signaling inhibition include neutrophils apoptosis and NETosis attenuation [21,22]. However, the bactericidal function of neutrophils under aberrant Rho-ROCK signaling activation is still poorly understood.

In this research, we found that the brains of mice suffered severe neutrophils infiltration after SEZ ATCC35246 infection. However, the bacterial burden (colony-forming units, CFU) in the brain was still numerous (10^8^–10^9^/organ) before death [23]. SEZ ATCC35246 presented neutrophil killing tolerance within the first 1 h in vitro. These data indicated the survival capacity of SEZ ATCC35246 against neutrophils killing at the initial engagement stage to neutrophils. This capacity showed a significant decrease when the *bifA* gene was knocked out while restored in complemental strain, which means *bifA* gene could play a role in SEZ ATCC35246 survival from neutrophils killing. Since BifA interacts with moesin in endothelium cells, we are curious about its interactive ability with other ERMs in neutrophils. Further experiments indicated that BifA interacted with ezrin and increased the phosphorylation of ezrin at Thr567 (which is located in the C-ERMAD and can lead to ezrin activation by phosphorylation) [10] in neutrophils. The p-ezrin activates Rac1 by recruiting Dbl, which could induce cytoskeleton derangement and disrupt the function of neutrophils. Collectively, this study reveals that BifA benefits SEZ survival from neutrophil killing by manipulating the Rho-ROCK pathway via the PCK-dependent ezrin phosphorylation.

## 2. Materials and Methods

### 2.1. Bacteria Strains, Growth Conditions, and Cell Culture

*Streptococcus equi* subsp. *zooepidemicus* ATCC35246 (SEZ ATCC35246) was isolated from a dead pig in Sichuan Province, China. The *bifA* gene deletion mutant ΔBif and the complement strain CBif were constructed in a previous study [7]. Neutrophil-like human leukemia cell line HL60 (ATCC^®^ CCL-240™) and HEK293T cells (ATCC^®^ CRL-3216™) were purchased from American Type Culture Collection (ATCC). Bacteria were cultured in Todd Hewitt Broth (THB) medium. The HL60 cells were maintained in IMDM medium with 20% fetal bovine serum (Gibco, Grand Island, NY, USA) in a 37 °C incubator containing 5% CO_2_. The HL60 cells were induced to differentiated HL60 (dHL60) cells by 1.3% dimethyl sulfoxide (DMSO) (Sigma-Aldrich, St. Louis, MO, USA) in IMDM for five days (Appendix A, see Appendix B) [24]. Primary mouse neutrophils were isolated from SPF ICR mice and kept in RPMI 1640 medium (Gibco, Grand Island, NY, USA) [25].

### 2.2. Ethics Statement and Animal Experiments

Four-week-old female BALB/c mice were purchased from the center of comparative medicine of Yangzhou University. Each experimental group of mice was injected with 1 × 10^6^ CFU of wild-type SEZ intravenously (*n* = 5), the bacteria were resuspended in Phosphate buffered saline (PBS, pH = 7.4). The control group was injected with the same volume PBS. All mice were sacrificed at 48 h after injection. Immunohistochemical staining was performed on the paraffin sections of mouse brain tissue; neutrophils were labeled with anti-Ly6c antibody (Abcam, Cambridge, MA, USA). Mouse neutrophils were isolated from bone marrow as previously described with a discontinuous Percoll density gradient centrifugation [25].

### 2.3. Protein Expression and Purification

Procedures for the expression of recombinant BifA have been described previously [7]. To purify GST-tagged BifA, BL21 (pLySs) harbors recombinant plasmid PGEX-6P-1-BifA was cultured to OD_600_ = 1.0 in LB medium at 37 °C, 180 rpm, then switched the cultural condition to 16 °C, 120 rpm with 1 mM IPTG for 16 h. Bacteria were collected by centrifugation and resuspended in lysis buffer (PBS, 140 mM NaCl, 2.7 mM KCl, 10 mM Na_2_HPO_4_, 1.8 mM KH_2_PO_4_, pH 7.3). After ultrasonic lysis, the soluble BifA protein was purified by GSTrap HP 5 mL column (GE Healthcare, Piscataway, NJ, USA). Precision protease and GSTrap FF 1 mL column (GE Healthcare) were used to remove GST-tagged. Sephadex 10/300 (GE Healthcare) was used for purification on the ÄKTA system (GE Healthcare).

### 2.4. Immunofluorescence and Confocal Microscopy

For the immunofluorescence (IF) assay, dHL60 cells were planted into a 15 mm glass-bottom cell culture dish (Corning, Corning, NY, USA) and treated with 10 μg/mL BifA protein for 2 h. Next, the cells were fixed with 4% paraformaldehyde (PFA) for 30 min after being rinsed twice with PBS. Fixed cells were permeabilized using 0.1% Triton X-100 and rinsed twice with PBS. The coverslips were blocked with blocking buffer (5% BSA and 0.1% Tween 20 in PBS) and incubated with mouse anti-BifA polyclonal antibody or rabbit anti-p-ezrin (T567) monoclonal antibody (Abcam) in blocking buffer for 1 h. For secondary antibodies, Alexa 594-conjugated goat anti-rabbit and Alexa 488-conjugated goat anti-mouse antibodies (Jackson Immunoresearch, West Grove, PA, USA) were diluted in blocking buffer. Finally, DAPI was used to stain the cell nuclei. A confocal laser scanning microscopy (Zeiss, LSM710, Jena, Thuringia, Germany) was used to acquire pictures from prepared samples. For densitometric analysis, pixel intensity was quantified using ImageJ software.

### 2.5. Co-Immunoprecipitation

Co-immunoprecipitation was used to examine the interaction between ezrin, truncated ezrin and BifA. The ezrin and truncated ezrin gene were separately amplified from HEK293T cells cDNA through PCR with primers listed below. The ezrin-F: 5′-ccaagcttctgcaggaattcatgccgaaaccaatcaatgtccg-3′ and ezrin-R: 5′-tatgggtatctagactcgagttacagggcctcgaactcgt-3′; ezrin-1-497aa-F: 5′-gcccaggcccgaattcatgccgaaaccaatcaatgtcc-3′ and ezrin-1-497aa-R: 5′-tagccggtacctcgagcgtgggctctgcgc-3′; ezrin-298-586aa-F: 5′-gcccaggcccgaattcgacaccatcgaggtgcagc-3′ and ezrin-298-586aa-R: 5′-tagccggtacctcgagcagggcctcgaactcgt-3′; ezrin-470-586aa-F: 5′-gcccaggcccgaattccccccgcccccac-3′ and ezrin-470-586aa-R: 5′-tagccggtacctcgagcagggcctcgaactcgt-3′, then inserted into the pCMV-C-HA vector using ClonExpress Ultra One Step Cloning Kit (Vazyme Biotech Co., Nanjing, JS, China). The construction of the BifA-pAcGFP plasmid has been described previously [7]. Plasmids of BifA-pAcGFP and ezrin-pCMV-HA were co-transfected into HEK293T cells. Lysates were harvested 48 h later with RAPI and incubated with mouse anti-HA or anti-GFP monoclonal antibody (CMCTAG, Milwaukee, WI, USA) for 1 h; the mixture was incubated with protein G agarose overnight at 4 °C. Beads were collected by centrifugation, washed with PBS, and then boiled with protein loading buffer for 5 min before western blot analysis.

For immunoprecipitation assay, dHL60 cells in different treatment groups were lysed with RIPA, anti-p-ezrin (T567) monoclonal antibody, anti-ezrin monoclonal antibody, and anti-Dbl monoclonal antibody (Santa Cruz Biotechnology, Dallas, TX, USA) were used for immunoprecipitation respectively.

### 2.6. Western Blot

Whole-cell lysate and immunoprecipitated supernatant were subjected to SDS-PAGE and transferred to PVDF membranes (Roch, Basel, Switzerland) using a semi-dry transfer apparatus (Bio-Rad, Hercules, CA, USA). Membranes were blocked with 10% milk powder in TBST (TBS and 0.1% Tween-20) for 2 h at room temperature. Membranes were probed with appropriate primary antibodies overnight at 4 °C, then washed and incubated for 1 h at room temperature with HRP conjugated secondary antibody. Chemiluminescence was induced by ECL reagent (Thermo, Waltham, MA, USA) and recorded on the ChemiDoc system (Bio-Rad, Hercules, CA, USA). Gray intensity was quantified with Image J. Primary antibodies used in this study included anti-HA (1:2000, CMCTAG), anti-GFP (1:2000, CMCTAG), anti-ezrin (1:1000, Abcam), anti-p-ezrin (T567) (1:1000, Abcam), anti-Rac1 (1:1000, Cytoskeleton, Denver, CO, USA), anti-GAPDH (1:2000, CMCTAG), and anti-Dbl antibodies (1:500, Santa Cruz Biotechnology). Secondary antibodies used in this study included goat anti-rabbit and goat anti-mouse IgG antibodies (ABGENT, San Diego, CA, USA).

### 2.7. Rac1 Activation Detection

According to the manufacturer’s guidelines, the activation of small GTPases was analyzed with the RhoA/Rac1/Cdc42 activation assay combo biochem kit (Cytoskeleton, Denver, CO, USA). In brief, dHL60 cells were treated with 10 μg/mL BifA protein for 2 h. Subsequently, cells were harvested in ice-cold cell lysis buffer containing protease inhibitors. Lysates containing equal amounts of protein were incubated at 4 °C with PAK-PBD beads for 1 h to harvest Rac1-GTP, elution was analyzed by western blot using mouse anti-Rac1 antibody (1:500, Cytoskeleton).

The Rac1 activity related to the p-Ezrin and Dbl was detected by an in vitro exchange reaction and pull-down detection [19]. BifA treated dHL60 cells were lysed with RIPA buffer. Dbl antibody-conjugated beads were added to the lysate at the preclear step for the depletion of Dbl. For the in vitro exchange reaction, 1 µg of recombinant Rac1 protein (Abcam) was preloaded with GDP incubating in 100 µL of loading buffer (50 mM Tris-HCl, pH 7.5, 50 mM NaCl, 5 mM EDTA, 1 mM DTT, 1 mg/mL BSA and 10 μM GDP) for 20 min at 25 °C. After the incubation, the reaction was stopped with 100 µL stop exchange buffer (50 mM Tris-HCl, pH 7.5, 10 mM MgCl_2_ and 1 mM DTT), then diluted with 1.5 mL exchange buffer (50 mM Tris-HCl, pH 7.5, 200 μM GTPγS and 2 mM MgCl_2_) as Rac1-GDP solution. Five hundred microliter Rac1-GDP solution was transferred to a tube containing the Dbl depleted cell lysate and then incubated with gentle agitation at 25 °C for 20 min. The reaction was stopped by adding 0.1 volume of stop buffer (50 mM Tris-HCl, pH 7.5, and 60 mM MgCl_2_). The amount of Rho-GTP in the reaction solution was measured by a pull-down method based on specific binding to Rhotekin-RBD followed by immunoblot with an anti-Rac1 antibody.

### 2.8. dHL60 Cells and Primary Mouse Neutrophils Bactericidal Assay

Neutrophil-like cells killing assays for SEZ wild-type strain and mutant strain refer to previously described with some modifications [26]. Briefly, isolated primary mouse neutrophils or differentiated HL60 cells were seeded at a density of 10^6^ cells/mL in 24-well cell culture plates with corresponding mediums. SEZ wild-type, ΔBif strain and the complement strain CBif were grown overnight; cultures were diluted into fresh THB and grown to mid-log phase (OD_600_ = 0.6). Bacteria were pre-treated prior to neutrophil challenge by resuspension in fetal bovine serum for 30 min at 37 °C. Neutrophil-challenged bacteria were incubated at 37 °C in 5% CO_2._ The cells were harvested with sterile double-distilled water at different time points and spread to THB agar with serial dilutions. Bacteria were cultured at 37 °C overnight. The CFU burden at 0 min was used as the denominator in the calculation of survival percentage.

### 2.9. Statistical Analysis

Data are presented as the Mean ± SD. Statistical analyses are performed with the Student’s *t*-test or ANOVA. The *p*-value < 0.05 is considered statistic significant.

## 3. Results

### 3.1. The bifA Gene Defective SEZ Has a Lower Tolerance to Neutrophils Killing In Vitro

Our previous research found that after the blood-brain barrier penetration, SEZ ATCC35246 can reach an extremely high CFU burden at 24 h after being challenged [23]. At the same time point, the histological section of the brain showed severe neutrophils infiltration in meninges (Figure 1A), indicating that bacteria are highly likely to be resistant to the stressful environment of neutrophils infiltration in vivo. Thus, it can survive and reproduce numerous colonies in the brain. In vitro experiments showed that more than half of the bacteria survive from 60 min killing of primary mice neutrophils (Figure 1B). Meanwhile, this survival capacity was significantly suppressed in the ΔBif strain with *bifA* gene defective within the first-hour engagement to primary mice neutrophils (Figure 1B) and dHL60 cells (Figure 1C). The absence of *bifA* gene appears to account for the reduced capacity of ΔBif to survive in neutrophils since the complementation restores neutrophils killing tolerance and elevates above that of ΔBif. It suggested that the *bifA* gene may confer SEZ the capacity of survival against neutrophils.

### 3.2. BifA Can Interact with Ezrin in Neutrophils and Elevate the p-Ezrin Level in PKC-Dependent Manner

BifA has been identified as an important virulent factor in blood-brain barrier penetration of SEZ ATCC35246 via interacting with moesin in endothelium cells. However, it is still unknown whether BifA could bind to other ERM family proteins in other cell types. Since BifA was responsible for neutrophils killing tolerance in vitro, we localized the ezrin and exogenous BifA protein in neutrophils. Interestingly, in the BifA treated dHL60 cells, p-ezrin and BifA were co-localized in the cytoplasm (Figure 2A). Negative control (NC) was purified parallel with BifA but from BL21 harboring empty plasmid used as backbone vector for BifA protein expression. The NC-treated neutrophils did not have as luminous p-ezrin signal as the BifA-treated ones, indicating that the BifA treatment might increase the ezrin phosphorylation level in neutrophils. Further experiments supported this presumption. In the first 2 h after BifA addition in dHL60 cells, the p-ezrin level increased along with time. In contrast, NC-treated dHL60 cells had stable p-ezrin at a low level (Figure 2B). In addition, we detected the ratio of p-ezrin/total-ezrin in dHL60 under BifA treatment with the phos-tag assay. The ratio increased from <0.1 to 1, indicating most ezrin was phosphorylated within 2 h (Figure 2C). However, under the treatment of staurosporine (an ATP-competitive and non-selective protein kinases inhibitor), BifA-induced ezrin phosphorylation was utterly abrogated, suggesting that PKC played a vital role in ezrin phosphorylation elevation relevant to BifA (Figure 2D). With the Co-IP approach, we found that GFP-BifA and HA-ezrin fused proteins co-expressed in 293T cells precipitated together by the anti-GFP monoclonal antibody. Yet, GFP alone has no affinity to ezrin at all (Figure 2E). To identify the interaction domains of ezrin and BifA, we expressed the truncated ezrin in 293T cells. As expected, the C-ERMAD domain of ezrin was essential for its interaction with BifA (Figure 2F), similar to the interaction pattern of BifA and moesin. Together, these observations indicate that BifA can interact with ezrin in neutrophils and increase the p-ezrin level in a PKC-dependent manner. BifA could be a universal ERMs interaction partner to their C-ERMAD domain, at least the two most widely distributed and abundant ERMs, ezrin and moesin.

### 3.3. BifA Triggered p-Ezrin Elevation Recruited Dbl and Led to Rac1 Activation

Ezrin is an upstream molecule of the Rho-ROCK pathway, and ezrin phosphorylation will lead to Rho-ROCK activation. This pathway regulates the reorganization of the actin cytoskeleton and affects phagocytic appetite [27]. We monitored the level of small G-protein activation in BifA-treated dHL60 cells by immunoblot with an antibody that recognizes Rac1-GTP and found that Rac1 was significantly activated to GTP bound form. The activation happened within 15 min after the addition of BifA, and the GDP-bound inactive Rac1 kept on converting to GTP-bound active Rac1 in 2 h (Figure 3A). This conversion could be, at least partially, induced by Dbl, a guanine nucleotide exchange factor (GEF) recruited by p-ezrin [19]. As expected, Dbl can be immunoprecipitated with p-ezrin by an anti-p-ezrin antibody from BifA treated dHL60 cells, though there seem to be a slight cross-reaction between anti-ezrin antibody and ezrin, their affinity is very low and the band in immunoblot is almost invisible (Figure 3B). To investigate the role of Dbl/p-ezrin interaction in Rac1 activation, we performed a Rho guanine nucleotide exchange activity assay. In the Dbl depleted sample, most Dbl was absorbed by anti-Dbl antibody-conjugated beads and removed from the reaction system. The results showed that though there was a high level of p-ezrin in dHL60 cells lysate, if Dbl was depleted, the active form Rac1 would still significantly reduce (Figure 3C). To detect the cytoskeleton dynamic of dHL60 cells under BifA treatment, we stained the F-actin with red fluorescence. The signal intensity of F-actin shifted away from the nucleus and concentrated to uropod of neutrophils, suggesting the cytoskeleton dynamic was changed after BifA treatment (Figure 3D). These data indicate that the phosphorylated ezrin still keeps the property of binding with Dbl after BifA treatment in neutrophils. The consequence is Rac1 aberrant activation, which could cause neutrophils cytoskeleton derangement and functions disruption.

### 3.4. Rho-ROCK Pathway Inhibitor Y27632 Restores Neutrophils Killing Capability Reduced by BifA

Y27632 dihydrochloride is a selective ROCK inhibitor [28]. To protect dHL60 cells from BifA-induced aberrant Rho-ROCK activation, we used 1 nM Y27632 to keep Rho-ROCK pathway activation to a normal level. The appropriate concentration (1 nM) of Y27632 significantly improved neutrophils killing capability against wild-type SEZ ATCC34246 within the first 60 min (Figure 4A). However, the promotion of the decay effect, along with Y27632 concentration rise (Figure 4B), indicates that either exorbitant inhibition or activation of Rho-ROCK has an adverse impact on neutrophils functions. The BifA works as an indirect activator to the Rho-ROCK pathway through increasing ezrin phosphorylation level to recruit Dbl, although this process can be seized with the appropriate concentration of Y27632 by manipulating Rho-ROCK pathway, neutrophils killing capacity to SEZ is restored under this circumstance.

## 4. Discussion

This research is an extended study on Fic-domain protein BifA for its pathogenesis during SEZ infection. We found BifA had ERM family proteins binding capability (ezrin in this study), as well as PKC-dependent phosphorylation activity for ERMs. This virulence factor is distributed in hypervirulent SEZ strains specifically [2], and its target ERM proteins are essential for most host cells functions [10]. Thus, BifA may affect multiple types of cells to enhance SEZ invasion. Besides endothelium, neutrophils were identified as another type of cells that could be affected by BifA. In this study, we investigated the downstream signal pathway of BifA/ezrin interaction and illustrated the Dbl dependent Rho-ROCK activation was the vital step in BifA pathogenesis (Figure 5).

Pathogen recognition and the subsequent recruitment of neutrophils to sites of infection are critical elements of the host defense against bacterial disease. After entering into the brain, SEZ infection led to severe neutrophils infiltration. However, instead of being eliminated by numerous neutrophils, the bacterial burden increased to a very high level in the brain [23]. BifA defective mutant (ΔBif) has a lower capability of survival against neutrophils killing in vitro. Phagocytosis is one of three major processes of neutrophils that is responsible for eliminating pathogens. It usually happens ahead of degranulation and neutrophil extracellular traps (NETs) that accompany cell death [29]. The survival capacity between WT and ΔBif showed significant difference only within the first 1 h in vitro. We postulated that BifA tended to affect the initial phagocytosis phase of neutrophils killing. However, as long as these cell death-dependent bactericidal functions were activated, reactive oxygen species (ROS) and other cytoplasmic granules contents released by neutrophils can still eliminate SEZ efficiently in vitro, further data will be necessary to reveal the specific function that has been affected by BifA during reducing neutrophils killing capability to SEZ.

As a Fic domain-containing protein, BifA had been identified to usurp moesin-dependent signaling and confer the BBB penetration capability to SEZ ATCC35246 [7], raising the question of the possible affinity of BifA to other ERM family proteins. The ezrin is a critical regulator of cytoskeletal-plasma membrane interactions [30]. Though the leucocytes were also able to express trace amounts of moesin and radixin [31], we can only detect ezrin as the primary ERM family protein in dHL60 cells with immunoblot. The phagocytosis and motility of neutrophils are relevant to normal ezrin phosphorylation. However, the aberrant ezrin phosphorylation level elevation is harmful, which has been identified as a consequence of BifA treatment in this research. Our data showed that BifA has the ability of binding ezrin to increase and keep its phosphorylation to an aberrant level, consistent with the phenotype of BifA and moesin interaction [7]. In addition, we found that the downstream of p-ezrin was Rho-ROCK signal transduction, and its activation relied on Dbl, a GDP/GTP exchange protein (GEP). We have not illustrated how BifA induces ERMs phosphorylation yet. Similar research about *Salmonella* Typhimurium indicated that its multidrug resistance-associated protein 2 (MRP2) induced ezrin activation via a PKC-α dependent pathway [32]. Though we have no direct evidence to support that the BifA-induced p-ezrin elevation was due to PKC-α, the PKC inhibitor staurosporine abrogated ezrin phosphorylation completely under BifA treatment, which could be considered as even indirect evidence. The research about Dbl recruitment by p-ezrin and Rho-ROCK activation promoted our knowledge about BifA function in SEZ infection through interaction with ERMs and indicated that the BifA-induced ezrin phosphorylation did not disturb its ability of Dbl recruitment [33].

Like many other actin-dependent processes, small G proteins play a significant role in regulating phagocytosis [34]. Phagocytosis contains multi-step cellular processes. The actin polymerization is a requisite for F-actin-driven engulfment. It provides the cytoskeletal framework to advance the plasma membrane of neutrophils over the bacteria and sequester them in phagosomes prior to killing [35]. The abnormal activation of a small G protein will make cytoskeletal dynamics collapse [36] and lead to phagocytosis dysfunction. Thus, if BifA increases the p-ezrin level and induces Rac1 activation extraordinarily, as we have observed in dHL60 cells, then it is likely that the phagocytosis and other actin-dependent processes of these cells would be impaired. The ROCK inhibitor Y27632 efficiently restored neutrophils killing function within the first 1 h, suggesting the feasibility of using this drug for preventing severe SEZ infection under emergency situation.

## Figures and Tables

**Figure 1 biomedicines-10-00932-f001:**
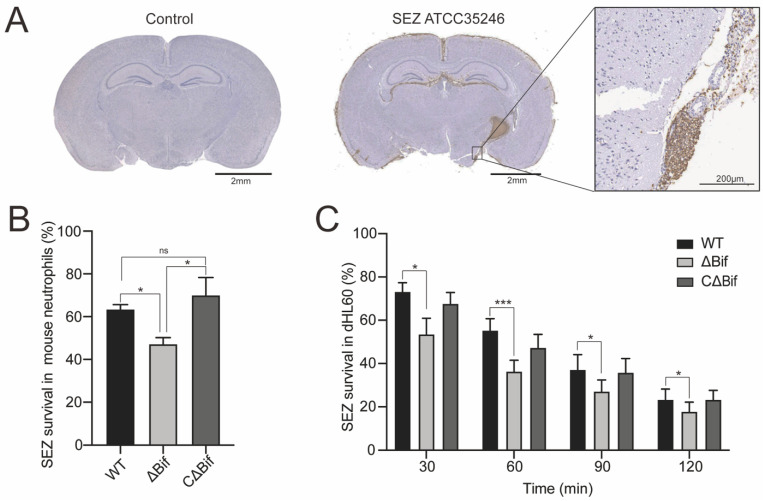
The *bifA* gene defective SEZ strain is more susceptible to neutrophils killing. (**A**) SPF BALB/c mice were challenged with 1 × 10^6^ CFU wild-type SEZ (SEZ ATCC 35246) or PBS as a control. Brains were stained with immunohistochemistry after the paraffin section. Granulocytes, especially the neutrophils, were stained densely. Severe neutrophils infiltration happened in SEZ infected mouse brain. (**B**) The survival percentage of SEZ, ΔBif and CΔBif after engaging with primary mouse neutrophils (multiplicity of infection, MOI = 1:10) for 60 min in vitro (*n* = 3, *p* values were calculated with one-way ANOVA, * indicates *p* < 0.05; ns indicates no significant difference) (**C**) The survival percentage of SEZ, ΔBif and CΔBif in dHL60 cells at different time points (*n* = 6, *p* values were calculated with one-way ANOVA for each time point, * indicates *p* < 0.05; *** indicates *p* < 0.001).

**Figure 2 biomedicines-10-00932-f002:**
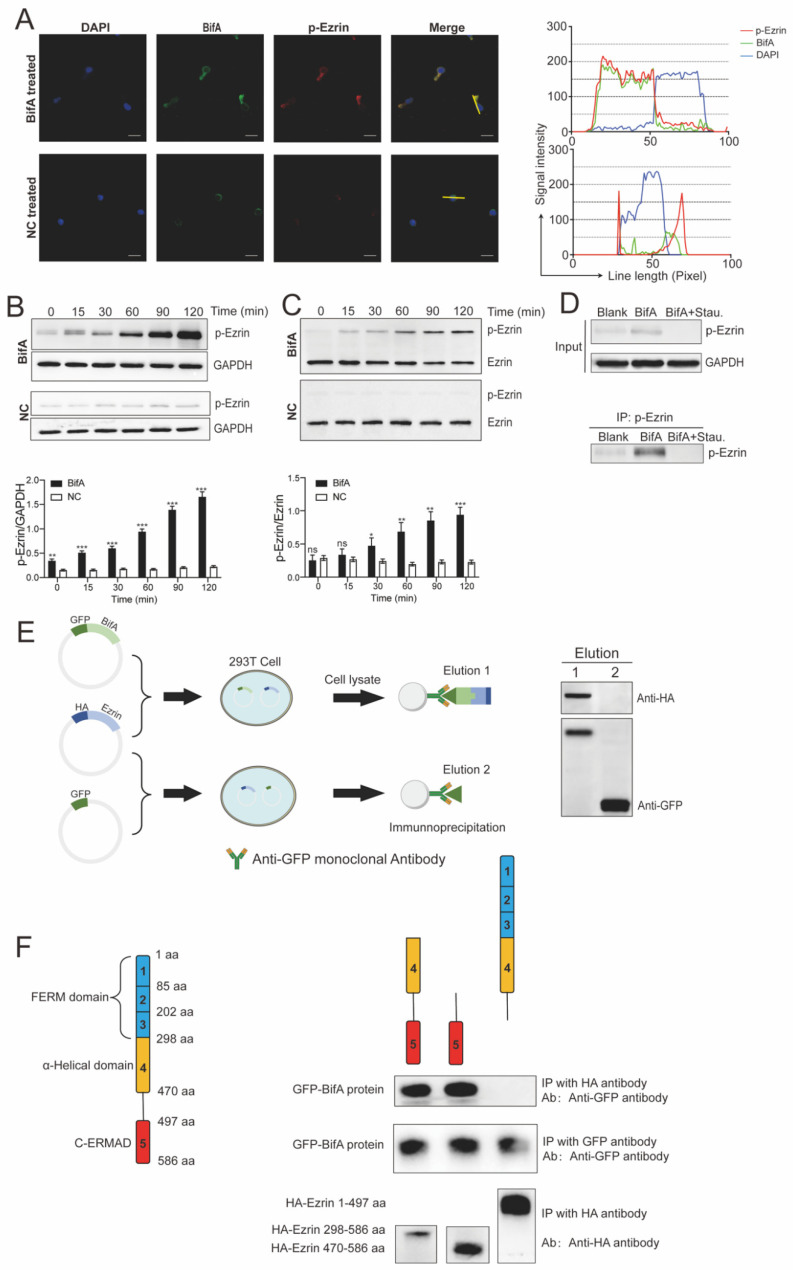
The *bifA* gene defective SEZ strain is more susceptible to neutrophils killing. BifA interacts with ezrin and elevates p-ezrin levels in dHL60 cells. (**A**) Observation of dHL60 cells incubated with BifA or Negative Control (NC) by immunofluorescence microscopy. BifA and p-ezrin were detected by immunostaining with an anti-BifA polyclonal antibody (green) and an anti-p-ezrin antibody (red), respectively, and DAPI was used as a counterstain for nuclei (blue). (Bar = 20 μm) The yellow lines in the merged pictures were in accordance with the line length (*X*-axis) of graphs on the right. The *Y*-axis showed the signal intensity of the three colors channels along with the yellow line. The overlap of green and red curves indicated the colocalization of BifA and p-ezrin in merged pictures. (**B**) Immunoblot detection of ezrin phosphorylation in dHL60 cells under BifA or NC treatment at different time points with anti-p-ezrin antibody. The gray intensity was measured with Image J. Gray intensity ratio of p-ezrin/GAPDH shown on the below graph (*n* = 3, ** indicates *p* < 0.01; *** indicates *p* < 0.001). (**C**) As the phos-tag can bind to p-ezrin and shift its mobility in gel, ezrin and p-ezrin were separated in phos-tag acrylamide gel, and detected by immunoblot. The gray intensity ratio of p-ezrin/GAPDH shown on the below graph (*n* = 3, * indicates *p* < 0.05; ** indicates *p* < 0.01; *** indicates *p* < 0.001; ns indicates no significant difference). (**D**) The phosphorylation level of ezrin in dHL60 cell lysate (input) after BifA or BifA + staurosporine (BifA + Stau.) treatment for 2 h was detected by immunoblot; the blank group was untreated cells. The p-ezrin was concentrated by immunoprecipitation with anti-p-ezrin antibody-conjugated beads and detected by immunoblot to confirm ezrin phosphorylation abolishment due to staurosporine. (**E**) The ezrin-HA-pCMV/BifA-pAcGFP or HA-pCMV/pAcGFP plasmids were co-transfected into 293T cells for GFP-BifA and HA-ezrin expression. The cell lysate was immunoprecipitated with an anti-GFP antibody. Elution 1 and 2 were detected by immunoblot with anti-HA or anti-GFP antibodies to identify the co-precipitation of GFP-BifA and HA-ezrin. (**F**) The truncated ezrin proteins with different domains were co-expressed with BifA protein in 293T cells and immunoprecipitated by anti-HA or anti-GPF antibody. Western blot was used to detect the precipitated proteins and verify their interaction.

**Figure 3 biomedicines-10-00932-f003:**
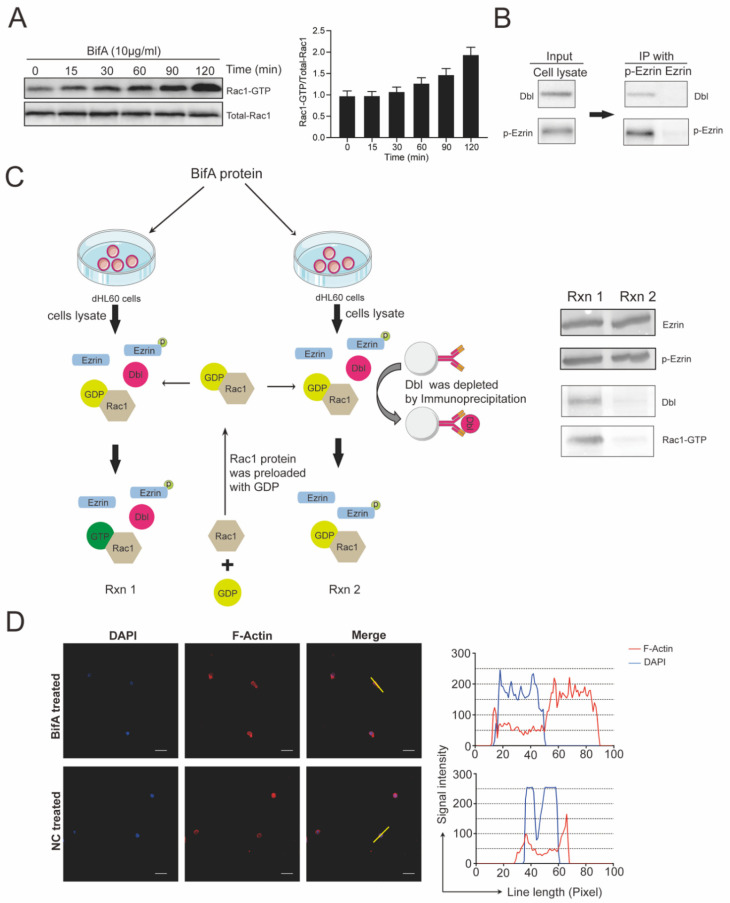
Rac1-GTP formation increased in BifA-treated dHL60 cells due to p-ezrin recruited Dbl. (**A**) Western blots were performed on lysates of dHL60 cells after BifA treatment. Total Rac1 and rhotekin protein precipitated GTP-bound Rac1 were detected with anti-Rac1 antibody. The graph shown below is normalized grayscale intensity analyses (measured with ImageJ software). (**B**) BifA-treated dHL60 cell lysate (input) was used to CO-IP with p-ezrin conjugated beads. Western blots were performed to detect the existence of Dbl and p-ezrin in the elution. (**C**) The dHL60 cell lysate was used as reaction system 1 (Rxn 1). Reaction system 2 (Rxn 2) was Dbl depleted dHL60 cell lysate generated with anti-Dbl antibody-conjugated beads for depletion. Ezrin, p-ezrin, Dbl in Rxn 1 and 2 were detected with immunoblot directly. The amount of Rho-GTP was measured by a pull-down method based on its specific binding to Rhotekin-RBD followed by immunoblot with an anti-Rac1 antibody. (**D**) Immunofluorescent staining of F-actin in dHL60 cells was observed by confocal microscopy. (Bar = 20 μm). The optical density of the scribed line in the picture was measured with Image J.

**Figure 4 biomedicines-10-00932-f004:**
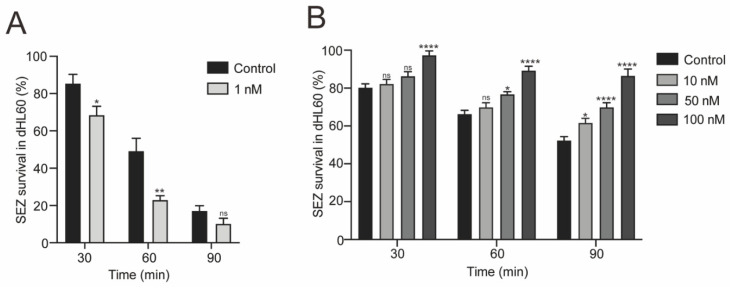
Appropriate concentration of Y27632 restored the dHL60 cells killing capability to SEZ. (**A**) The survival percentage of wild-type SEZ in dHL60 cells at different time points. With 1 nM Y27632, dHL60 killed more bacteria within 60 min than control group (*n* = 3, *p* values were calculated with Student’s *t*-test for each time point, * indicates *p* < 0.05; ** indicates *p* < 0.01; ns indicates no significant difference compare to control.). (**B**) In contrast, high concentration (10 nM, 50 nM and 100 nM) of Y27632 impaired dHL60 killing capability to wild-type SEZ. (*n* = 3, *p* values were calculated with one-way ANOVA for each time point, * indicates *p* < 0.05; **** indicates *p* < 0.0001; ns indicates no significant difference compare to control).

**Figure 5 biomedicines-10-00932-f005:**
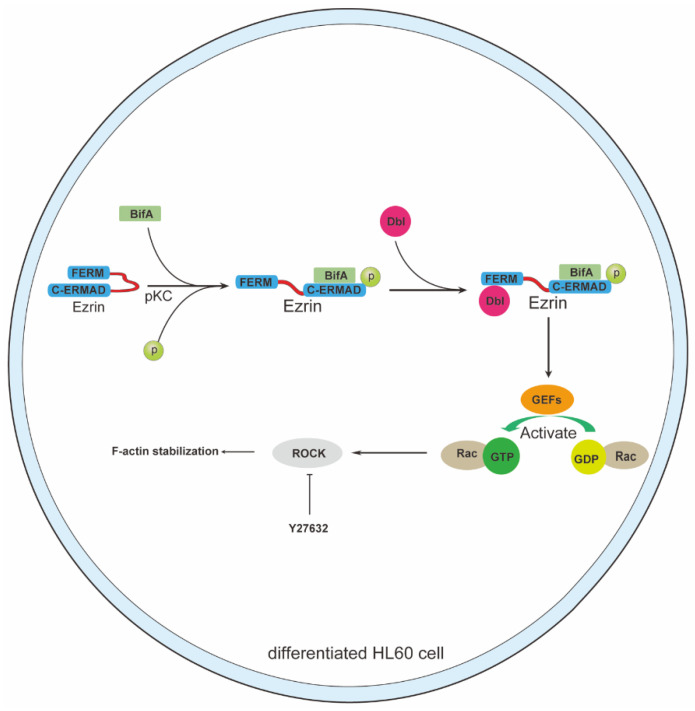
Schematic model of molecular mechanism of BifA decreases neutrophils killing capability to SEZ. BifA increases and keeps the ezrin aberrant phosphorylation level in a PKC-dependent manner. The p-ezrin recruits Dbl (guanine nucleotide-exchange factor) and promotes inactive GDP-bound Rac1 to transfer to active GTP-bound Rac1. The extraordinary activation of Rac1 may lead to phagocytosis dysfunction and impair other actin-dependent processes of dHL60 cells. The appropriate concentration of ROCK inhibitor Y27632 restores neutrophils function by maintaining Rho-ROCK pathway stability.

## Data Availability

The data that supports the findings of this study is available from the corresponding author upon a reasonable request.

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
