# Peer review of "BifA Triggers Phosphorylation of Ezrin to Benefit Streptococcus equi subsp. zooepidemicus Survival from Neutrophils Killing"

_biomedicines, 2022, doi:10.3390/biomedicines10050932_

Round 1
Reviewer 1 Report
In this research manuscript entitled ‘BifA triggers phosphorylation of ezrin to benefit Streptococcus equi subsp. zooepidemicus survival from neutrophils killing’ Pan & al. investigated the role of BifA protein in the neutrophils resistance in Streptococcus equi subsp. zooepidemicus (SEZ). Authors show that BifA protein increases tolerance to neutrophils killing in an in vitro model using mouse neutrophils and differentiated dHL60 cells. Interestingly, BifA seems to partially protect SEZ from neutrophils killing at 30 and 60 min. Authors hypothesized that BifA may interact with ERM family proteins (for ezrin, radixin, moesin, ERMs). In this paper, the authors show that BifA indeed interacts with the C-ERMAD domain of erzin and phosphorylate erzin by co-immunoprecipitation and western blotting analysis using p-erzin antibody (T567 residue). Finally, the authors show that BifA-mediated ezrin phosphorylation will lead to Rho-ROCK activation that may lead to neutrophils phagocytosis function reduction.
I don't have major concerns about this manuscript. Overall I think the manuscript is well written and introduce interesting result. However, I think some major/minor points need to be corrected or addressed in the introduction/discussion.
Major point:
-The authors use p-erzin monoclonal antibody to recognize phosphorylated T567 residue.
Q1. why directly use this exact residue?
Q2.On Abcam product sheet it specify (erzin (pThr567), Radixin (pThr564), Moesin (pThr558) was used as the immunogen. Can p-Radixin and p-Moesin be also detected in WB experiment? Please specify or discuss.
-line 186. Bacteria were opsonized before in vitro infection. Opsonisation helps phagocytosis and killing by neutrophil. Could BifA decrease opsonization? Please bring supportive information or additional experiment.
-Statistical analysis: for student T-test distribution should follow a normal distribution, is it the case here? Also, for group analysis, ANOVA should be used.
-Fig 4A. Why authors didn’t include delta-bifA mutant to compare with Y27632 treatment.
Minor point:
-gene and protein nomenclature need to be reviewed across the manuscript: gene should be in italic with no capital letter (bifA) whereas protein needs no italic and a capital letter (BifA). In vitro should also be in italic. Salmonella Typhimurium as well (line 370)
-The authors should introduce a little more the BifA protein; What is a Fic domain, what is the function of Fic domain, BifA is an extracellular or membrane protein? How it is delivered into the host cell (secretion system) ?
-Some references need to be added; lines 51-52. Line 350. Line 367
-line 71-72; ‘dramatic’ is subjective, I would rather remove it.
-line 99; Bacteria were resuspended in which buffer or medium before inoculation?
-line 218. C) what is the MOI?
-line 243: M&M about truncated construction?
-Figs 2A and 3D, ‘signal intensity’ not singal
-Fig 3A. The authors used a concentration of 10ug/ml. This concentration is based on what information? 10ug/ml seems high, do you expect bacteria to produce such amount in cells? Did the authors try other concentrations?
-line 346: Viral? Or vital
Reviewer 2 Report
This is an interesting article on the interaction of BifA with Ezrin in granulocytes infected with Streptococcus equi subsp. zooepidemicus. The manuscript is demonstrating that BifA is not only capable of interacting with moesin, but also with Ezrin, and this leads to the aberrant activation of the Rho-ROCK pathway. I have no major concerns with regard to the quality of the research work done here. However, the English language should be extensively revised by a native speaker. In addition, some figures have very small labelling, in particular Figs. 2E, 3C and 5. Finally, the species names in the bibliography are not written in italics.
